# On Adversarial Training without Perturbing all Examples

## Abstract

Adversarial Training (AT) is the de-facto standard for improving robustness against adversarial examples. This usually involves a multi-step adversarial attack applied on each example during training. In this paper, we explore only constructing Adversarial Examples (AEs) on a subset of the training examples. That is, we split the training set in two subsets $A$ and $B$, train models on both ($A \cup B$) but construct AEs only for examples in $A$. Starting with $A$ containing only a single class, we systematically increase the size of $A$ and consider splitting by class and by examples. We observe that: (i) adv. robustness transfers by difficulty and to classes in $B$ that have never been adv. attacked during training, (ii) we observe a tendency for hard examples to provide better robustness transfer than easy examples, yet find this tendency to diminish with increasing complexity of datasets (iii) generating AEs on only $50\%$ of training data is sufficient to recover most of the baseline AT performance even on ImageNet. We observe similar transfer properties across tasks, where generating AEs on only $30\%$ of data can recover baseline robustness on the target task. We evaluate our subset analysis on a wide variety of image datasets like CIFAR-10, CIFAR-100, ImageNet-200 and show transfer to SVHN, Oxford-Flowers-102 and Caltech-256. In contrast to conventional practice, our experiments indicate that the utility of computing AEs varies by class and examples and that weighting examples from $A$ higher than $B$ provides high transfer performance.

## 1 Introduction

Imperceptible changes in the input can change the output of a well performing model dramatically. These so-called Adversarial Examples (AEs) have been the focus of a large body on deep learning vulnerabilities of works since its discovery [1]. To date, Adversarial Training (AT) [2, 3] and its variants [4–6] is the de-facto state-of-the-art in improving the robustness against AEs. Essentially, AT generates adversarial perturbations for all examples seen during training. While adversarial training is known to transfer robustness to downstream tasks [7–9] and that robustness is distributed unevenly across classes [10, 11], common practice dictates that AT "sees" adversarial examples corresponding to the whole training data, including all classes and concepts therein. This is independent of whether only adversarial robustness is optimized or a trade-off between robustness and clean performance is desired [12]. This also holds for variants that treat individual examples differently [13–15] or adaptively select subsets to attack during training to reduce computational overhead [16, 17]. It is largely unclear how adversarial robustness is affected when training is limited to seeing adversarial examples only on specific subsets of the training data.

To shed light on this issue, we consider the adversarial training setup depicted in figure 1, called Subset Adversarial Training (SAT), where we split the training data into two subsets $A$ and $B$, train the model conventionally on the union ($A \cup B$), but generate AEs only on examples from $A$ (indicated

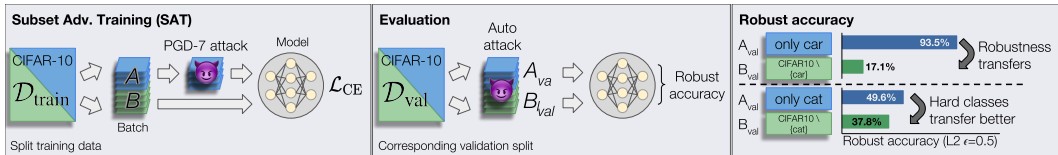

Figure 1: Adversarial robustness transfers among classes. Using Subset Adversarial Training (SAT), during which only a subset of all training examples ($A$) are attacked, we show that robust training even on a single class provides robustness transfer to all other, non adv. trained, classes ($B$). E.g., SAT for $A$=cat, we observe an robust accuracy of 37.8% on $B$. Noteworthy is the difference of transfer utility between classes. I.e. $A$=car provides very little transfer to $B$ (17.1%). We investigate this transfer among classes and provide new insights for robustness transfer to downstream tasks.

by the emoji). For example, we can split training data by class, with $A = \{car\}$ or $A = \{cat\}$ and $B = A^c$, and investigate how adversarial robustness transfers. Surprisingly, we observe significant adversarial robustness on $B_{\text{val}}$ at test time, the degree of which depends on the class(es) in $A$. Of course, $A$ and $B$ can be arbitrary partitions of the training data. For example, we could put only "difficult" examples in $A$ during training. At test time, we evaluate overall adversarial robustness (since there is no natural split into $A_{\text{val}}$ or $B_{\text{val}}$). These experiments reveal a rather complex interaction of adversarial robustness between classes and examples.

Our analysis provides a set of **contributions** revealing a surprising generalizability of robustness towards non-adv. trained classes and examples even under scarce training data setups. **First**, selecting subsets of whole classes, we find that SAT provides transfer of adversarial robustness to classes which have never been attacked during training. E.g. only generating adversarial examples for class *car* on CIFAR-10, achieves a non-trivial robust accuracy of 17.1% on all remaining CIFAR-10 classes (see figure 1, right). **Secondly**, we observe classes and examples that are hard to classify do generally provide better robustness transfer than easier ones. I.e. class *cat* achieves more than twice the robust accuracy on the remaining classes (37.8%) over class *car* (17.1%). **Thirdly**, SAT with 50% of training data is sufficient to recover the baseline performance with vanilla AT even on hard datasets like ImageNet. Lastly, we observe similar transfer properties of SATed models to downstream tasks. In this setting, exposing the model to only 30% of AEs during training, can recover baseline AT performance on the target task.

## 2   Related Work

Since their discovery [1], robustness against adversarial examples has mainly been tackled using adversarial training [18, 2, 4]. Among many others, prior work proposed adversarial training variants working with example-dependent threat models [19, 13–15], acknowledging that examples can have different difficulties. Some works also mine hard examples [16] or progressively prune a portion of the training examples throughout training [17, 20]. However, all of these methods generally assume access to adversarial examples on the whole training set. That is, while individual examples can be dropped during training or are treated depending on difficulty, the model can see adversarial perturbations for these examples if deemed necessary. Adversarial training is also known to transfer robustness to downstream tasks [8, 9, 7] and adversarially robust representations can be learned in a self-supervised fashion [21]. Here, a robust backbone is often adapted to the target task by re-training a shallow classifier – sometimes in an adversarial fashion. It is generally not studied whether seeing adversarial examples on the whole training set is required for good transfer. This is despite evidence that achieving adversarial robustness is easier for some classes/concepts than for others [22, 23, 11, 10], also for robustness transfer [24]. Complementing these works, we consider only constructing adversarial examples on a pre-defined subset of the training set, not informed by the model or training procedure, and study how robustness transfers across examples and tasks.

## 3  Background and Method

### 3.1  Adversarial Training (AT)

It is a well known fact that conventional deep networks are vulnerable to small, often imperceptible, changes in the input. As mitigation, AT is a common approach to extend the empirical risk minimization framework [2]. Let $(x, y) \in \mathcal{D}_{\text{train}}$ be a training set of example and label pairs and $\theta$ be trainable parameters, then AT is defined as:

$$\min_{\theta} \mathbb{E}_{(x,y) \in \mathcal{D}_{\text{train}}} \left[ \max_{||\delta||_2 \leq \epsilon} \mathcal{L}(x + \delta, y; \theta) \right], \tag{1}$$

where $\delta$ is a perturbation that maximizes the training loss $\mathcal{L}$ and thus training error. The idea being that, simultaneously to minimizing the training loss, the loss is also optimized to be stable within a small space $\epsilon$ around each training example $||\delta||_2 \leq \epsilon$ (we consider the $L_2$ norm). This additional inner maximization is solved by an iterative loop; conventionally consisting of 7 or more steps. In some settings [18, 12, 4], the robust loss is combined with the corresponding loss on clean examples in a weighted fashion to control the trade-off between adversarial robustness and clean performance.

### 3.2  AT without Perturbing all Training Examples

Most proposed AT methodologies generate AEs on the whole training set. This being also valid for methods which adaptively select subsets [16, 17] during training or more traditional AT in which only a subset per batch is adversarially attacked. These methods do not guarantee the exclusion of examples, that is, the model is likely to see an AE for every example in the training set. From a broader perspective, the necessity to generate AEs exhaustively for all classes appears unfortunate though. Ideally, we desire robust models to be scalable, i.e. transfer flexibly from few examples and across classes to unseen ones [25]. We propose SAT to investigate to what extent AT provides this utility. To formalize, let $A$ be a training subset and $B$ contain the complement: $A \subset \mathcal{D}_{\text{train}}, B = \mathcal{D}_{\text{train}} \setminus A$. Then SAT applies the inner maximization loop of AT on the subset $A$ only; on $B$ the conventional empirical risk is minimized:

$$\min_{\theta} \mathbb{E}_{(x,y) \in \mathcal{D}_{\text{train}}} \left[ w_A \mathbb{1}_{(x,y) \in A} \max_{||\delta||_2 \leq \epsilon} \mathcal{L}(x + \delta, y; \theta) + w_B \mathbb{1}_{(x,y) \in B} \mathcal{L}(x, y; \theta) \right], \tag{2}$$

where $\mathbb{1}_{(x,y) \in A}$ is 1 when the training example is in $A$ and 0 otherwise. $w_A$ and $w_B$ define optional weights, which are by default both set to 1. Note that this is different from balancing robust and clean loss as discussed in [18, 12, 4], where the model still encounters adversarial examples on the whole training set.

**Loss balancing.** The formulation in equation 2 implies an imbalance between left and right loss as soon as the training split is not even ($|A| \neq |B|$). To counteract, we assign different values to $w_A$ and $w_B$ based on their subset size. E.g., to equalize the loss between both subsets, we assign $w_B = 1$ and $w_A = |B|/|A|$. We will utilize this loss balancing to improve robustness for transfer learning in section 4.3.

### 3.3  Training and evaluation recipes

Consider the depiction of SAT in figure 1. Prior to training, the training set is split into $A$ and $B$ (left). For evaluation (middle), we split the validation set into a corresponding split of $A_{\text{val}}$ and $B_{\text{val}}$, if possible. For **Class-subset Adversarial Training (CSAT)**, this split aligns with the classes on the dataset: $A$ and $B$ are all training examples corresponding to two disjoint sets of classes while $A_{\text{val}}$ and $B_{\text{val}}$ are the corresponding test examples of these classes. As experimenting with all possible splits of classes is infeasible, we motivate splits by class difficulty where we measure difficulty by the average entropy of predictions per class – introduced as $\mathcal{H}_C$ in the next paragraph.  In contrast, we can also split based on individual example difficulty.  We provide empirical support for this approach in the experimental section 4. Additionally, example difficulty has been frequently linked to proximity between decision boundary and example [26, 13, 15, 16, 27]. The closer the example is to the boundary, the harder it is likely to classify. The hypothesis: hard examples provide a larger

contribution to training robust models, since they optimize for large margins [13, 14]. We refer to this experiment as **Example-subset Adversarial Training (ESAT)**. In contrast to CSAT, however, there is no natural split of the test examples into $A_{\text{val}}$ and $B_{\text{val}}$ such that we evaluate robustness on the whole test set (i.e., $\mathcal{D}_{\text{val}}$).

As difficulty metric, we utilize entropy over softmax, which we empirically find to be as suitable as alternative metrics (discussed in the supplement). Consider a training set example $x \in \mathcal{D}_{train}$ and a classifier $f$ mapping from input space to logit space with $N$ logits. Then the entropy of example $x$ is determined by $\mathcal{H}(f(x))$ and of a whole class $C \subset \mathcal{D}_{train}$ is determined by $\mathcal{H}_C(f)$ – the average over all examples in $C$:

$$\mathcal{H}(f(x)) = -\sum_{i=1}^{N} \sigma_i(f(x)) \cdot \log \sigma_i(f(x)), \quad \mathcal{H}_C(f) = \frac{1}{|C|} \sum_{x \in C} \mathcal{H}(f(x)),$$

where $\sigma$ denotes the softmax function. For our SAT setting, we rank examples prior to adversarial training. This requires a classifier pretrained on $\mathcal{D}_{train}$ enabling the calculation of the entropy. To strictly separate the effects between entropy and AT, we determine the entropy using a non-robust classifier trained without AT. Similar to [27], we aggregate the classifier states at multiple epochs during training and average the entropies. Let $f_1, f_2, ...f_M$ be snapshots of the classifier from multiple epochs during training, where $M$ denotes the number of training epochs. Then the average entropy for an example is given by $\overline{\mathcal{H}}(x)$ and for a class by $\overline{\mathcal{H}}_C(f)$:

$$\overline{\mathcal{H}}(x) = \frac{1}{M} \sum_{e=1}^{M} \mathcal{H}(f_e(x)), \quad \overline{\mathcal{H}}_C = \frac{1}{M} \sum_{e=1}^{M} \mathcal{H}_C(f_e). \tag{3}$$

# 4 Experiments

As aforementioned, common practice performs AT for the whole training set. In the following, we explore CSAT and ESAT, which splits the training set in two subsets $A$ and $B$ and only constructs AEs for $A$ such that the model never sees AEs for $B$. We start with single-class CSAT – $A$ contains only examples of a single class – and increase the size of $A$ (section 4.1) by utilizing the entropy ranking of classes $\mathcal{H}_C$ (equation 3). ESAT, which splits into example subsets is discussed in section 4.2. Both SAT variants reveal complex interactions between classes and examples while indicating that few AEs can provide high transfer performance to downstream tasks when weighted appropriately (section 4.3).

**Training and evaluation details.** Since AT is prone to overfitting [28], it is common practice to stop training when robust accuracy on a hold-out set is at its peak. This typically happens after a learning rate decay. We adopt this "early stopping" for all our experiments by following the methodology in [28] but utilize Auto Attack (AA) to evaluate robust accuracy. Throughout the course of the training, we evaluate AA on $10\%$ of the validation data $\mathcal{D}_{\text{val}}$ after each learning rate decay and perform final evaluation with the model providing the highest robust accuracy. This final evaluation is performed on the remaining $90\%$ of validation data. This AA split is fixed throughout experiments to provide consistency. If not specified otherwise, we generate adversarial examples during training with PGD-7 within an $L_2$ epsilon ball of $\epsilon = 0.5$ (all CIFAR variants) or $\epsilon = 3.0$ (all ImageNet variants) – typical configurations found in related work. We train all models from scratch and use ResNet-18 [29] for all CIFAR-10 and CIFAR-100 [30] experiments and ResNet-50 for all ImageNet-200 experiments. Here, ImageNet-200 corresponds to the ImageNet-A subset [31] to render random baseline experiments tractable (to reduce training time). This ImageNet-200 dataset, contains 200 classes that retain the class variety and breadth of regular ImageNet, but remove classes that are similar to each other (e.g. fine-grained dog types). We use all training and validation examples from ImageNet [32] that correspond to this subset classes. All training details can be found in the supplement.

## 4.1 Class subset splits

We start by investigating the interactions between individual classes in $A$ using CSAT on CIFAR-10, followed by an investigation on increasing the number of classes. **Single-class subsets (CSAT).** We train all possible, single class CSAT runs (10) and evaluate robust accuracies on the adv. trained class (A) and the non-adv. trained classes (B). The results are shown in figure 2, left. Each rows

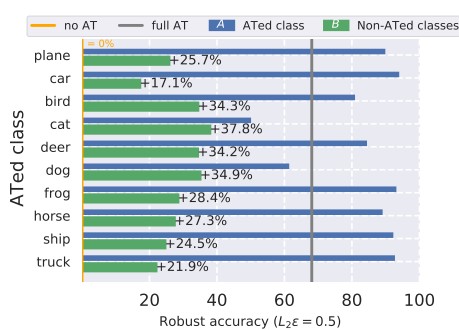
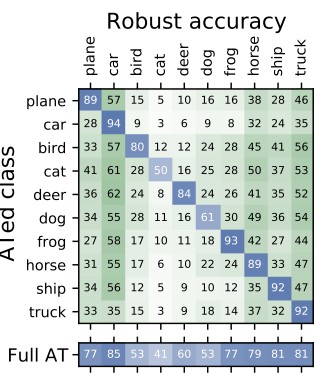

Figure 2: CSAT on a single CIFAR-10 class $A$ (blue), we observe non-trivial transfer to the non-adv. trained classes $B$ (green). Classes considered hard in CIFAR-10 (cat) offer best generalization (+37.8% gain on non-adv. trained), while easy classes offer the worst (car, +17.1% gained). Note that without AT, robust accuracy is close to 0% for all classes (orange). Right: same as left, but robust accuracy is evaluated per class (along columns). Here, we observe an unexpected transfer property: hard classes provide better transfer to seemingly unrelated classes (cat → truck: 53%) than related classes (car → truck: 35%).

represents a different training run. Note that the baseline robust accuracy, trained without AT achieves practically 0% (indicated by red line). Most importantly, we observe non-trivial robustness gains for all classes that have never been attack during training ($B$-sets). That is, irrespective of the chosen class, we gain at least 17.1% robust accuracy (A=car) on the remaining classes and can gain up to 37.8% robust accuracy when A=cat. These robustness gains are unexpectedly good, given many features of the non-adv. trained classes can be assumed to not be trained robustly.

To investigate this phenomenon further, we analyze robust gains for each individual class and present robust accuracies in the matrix in figure 2, right, where training runs are listed in rows and robust accuracies per class are listed in columns. Blue cells denote the adv. trained class and green cells denote non-adv. trained classes. While we see some expected transfer properties, e.g. CSAT on *car* provides greater robust accuracy on the related class *truck* (46%) than unrelated animal classes *bird, cat, deer, dog* (between 5% and 16%), the reverse is not straight-forward. CSAT on *bird* provides 56% robust accuracy on the seemingly unrelated class *truck*, 10%-points more than CSAT on *car*. More generally, animal classes provide stronger robustness throughout all classes than inanimate classes. We observe, that these classes are also harder to classify and have a higher entropy $\overline{\mathcal{H}}_C$ as shown in figure 3.

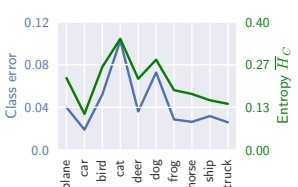

Figure 3: The hardest classes (blue) have the highest entropy (green).

**Many-class subsets (CSAT).** To increase the number of classes in $A$ while maintaining a minimal computational complexity, we utilize the average class entropy $\overline{\mathcal{H}}_C$ proposed in equation 3 to inform us which ranking to select from. To improve clarity, we begin with a reduced set of experiments on CIFAR-10 before transitioning to larger datasets. We utilize the observed correlation between class difficulty, average class entropy and robustness transfer $\overline{H}_C$ to rank classes and construct 4 adv. trained subsets. Ranked by class entropy $\overline{\mathcal{H}}_C$, we select 4 subsets showing in figure 4, left. As observed before, *cat* and *dog* are hardest and thus first chosen to be in subset $A$. *Truck* and *car* on the other hand are easiest and thus last. To gauge the utility of this ranking, we provide a robust and clean accuracy comparison with a random baseline in figure 4, center and right. I.e., for each subset $A$ we select 10 random subsets and report mean and std. deviation (red line and shaded area). Similar to the single-class setup, we observe subsets of the hardest classes to consistently outperform the random baseline (upper middle plot), up until a subset size of $|A| = 8$, when it draws even. Also note that the robust accuracy on $B_{\text{val}}$ is improved across all splits, thus providing support that harder classes – as initially observed on animate vs inanimate classes – offer greater robustness transfer.

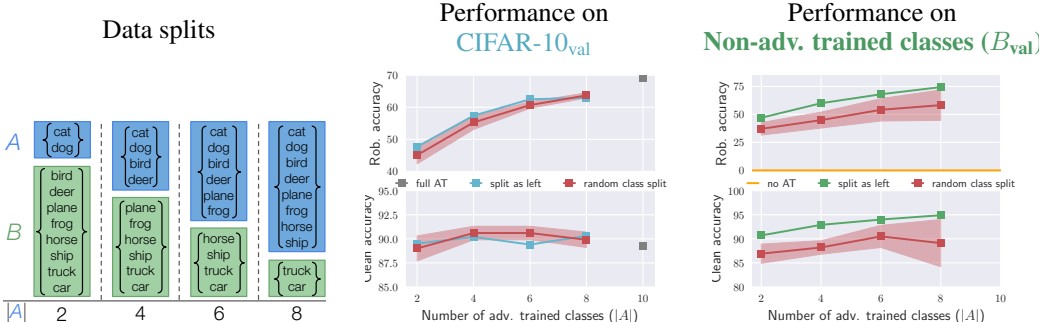

Figure 4: Ranking CIFAR10 classes by difficulty (using entropy as proxy), we perform CSAT with an increasing size of adv. trained classes in $A$. Class splits used for training ($A$ and $B$) are stated on the left. The resulting robust and clean accuracies on the validation set is shown on the right, separated into performance on $B_{\text{val}}$ and *all*. Compared with a random baseline of random class ranking (red), we find the ranking by difficulty to have consistently better transfer to non-adv. trained classes ($B$). Overall, this results in an improved robust accuracy on average over all classes.

For our experiments on larger datasets like CIFAR-100 and ImageNet-200, we additionally evaluate a third ranking strategy. Beside selecting at random and selecting the hardest first, we additionally compare with selecting the easiest (inverting the entropy ranking). We construct 9 subsets per type of ranking (instead of 4) and report robust accuracies for selecting the easiest classes as well. Results are presented in three columns in figure 5; one dataset per column. As before, we show robust accuracies on the tested dataset (upper row) and robust accuracies on $B_{\text{val}}$ (lower row). For CIFAR-10, we calculate mean and std. dev. over 10 runs, for CIFAR-100 over 5 runs and for ImageNet-200 over 3 runs. Selecting hardest first (highest entropy) is marked as a solid line and easiest first (lowest entropy) as a dashed line. First and foremost, we observe that irrespective of the dataset and the size of $A$, we see robustness transfer to $B_{\text{val}}$. This transfer remains greatest with classes we consider hard, while easy classes provide the least. Nonetheless, we see diminishing returns of such an informed ranking when dataset complexity is increased. E.g. the gap between dashed and solid line on ImageNet-200 is small and random class selection is on-par with the best. The results are similar on CIFAR-100, as shown in figure 5, middle). Based on these results, entropy ranking and selecting classes provides only slight improvements in general. Importantly though, we continue to see the tendency of increased robustness transfer to $B_{\text{val}}$, which we will come back to in section 4.3.

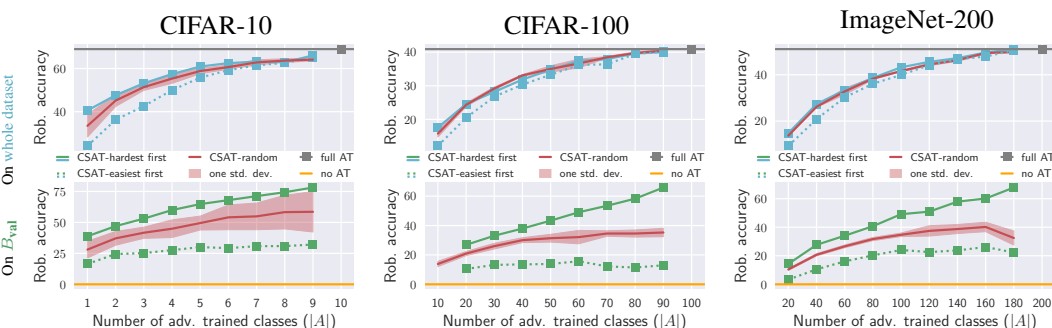

Figure 5: Class-subset Adversarial Training (CSAT) produces non-trivial robustness on classes that have never been attacked during training ($B_{\text{val}}$). Along the x-axes we increase the class subset size of $A$ on which AEs are constructed and compare three different class-selection strategies: select hardest first (solid lines), select easiest first (dashed line) and select at random (red). On average, random selection performs as well as informed ranking (upper row), while the robustness transfer to $B_{\text{val}}$ is best for the hardest classes (lower row). AT on a single class provides already much greater robust accuracies than without AT (orange).

## 4.2 Example subset splits (ESAT)

Considering that splits along classes are inefficient in terms of reaching the full potential of adversarial robustness, we investigate ranking examples across the whole dataset (ESAT). We follow with the same setup as before but rank examples – and not classes – by entropy $\overline{\mathcal{H}}$. Since it is not feasible to construct corresponding rankings on the validation set, we cannot gauge robustness transfer to $B_{\text{val}}$. Instead, we will test transfer performance to downstream tasks in section 4.3. We consequently report robust accuracy and clean accuracy on the whole validation set in figure 6.

Firstly, note that the increase in robust accuracy is more rapid than with CSAT w.r.t. the size of $A$. AT only on $50\%$ of training data ($25k$ examples on CIFAR and $112k$ on ImageNet-200) and the resulting average robust accuracy is very close to the baseline AT performance (gray line). Secondly, note that gap between hard (solid line) and easy example selection (dashed line) has substantially widened. In practice, it is therefore possible to accidentally select poor performing subsets, although the chance appears to be low given the narrow variance of random rankings (red). To some extent, this observation supports the hypothesis that examples far from the decision border (the easiest to classify) provide the least contribution to robustness gains. This is also supported by the reverse gap in clean accuracy (bottom row in figure 6). That is, easiest-first-selection results in higher clean accuracies than hardest-first, while robust accuracies are much lower. In contrast however, we observe random rankings (red) to achieve similar performances to hard rankings (solid lines) on all datasets and subset sizes. This is somewhat unexpected, especially on small sizes of $A$ (e.g. $5k$). Given the results, we conjecture that the proximity to the decision boundary plays a subordinate role to increasing robustness. Instead, it is plausible to assume that diversity in the training data has a large impact on learning robust features, also indicated by [33].

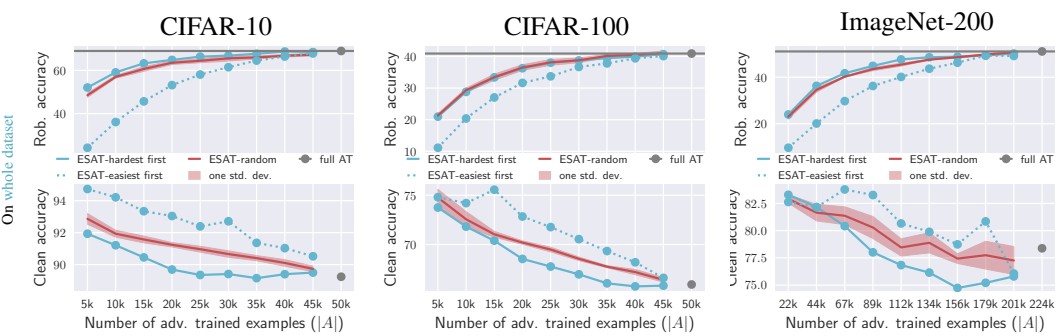

Figure 6: Example-subset Adversarial Training (ESAT) on CIFAR datasets and ImageNet-200, provide quick convergence to a full AT baseline (gray line and dot) with increasing size of $A$. We report robust accuracy (upper row) and clean accuracy (lower row) and observe similar characteristics as with CSAT (figure 5). I.e., selecting the hardest examples first (solid line) provide higher rob. accuracy than easy ones (dashed line), although the gap substantially widens. Random example selection (red) provides competitive performance on average. Across all datasets, we see the common clean accuracy decrease while robust accuracy increases [34].

## 4.3 Transfer to downstream tasks

Previous experiments on ESAT could not provide explicit robust accuracies on the non-adv. trained subset $B_{\text{val}}$ since training and testing splits do not align naturally – recall the evaluation recipe outlined in section 3.3. In order to test transfer performance regardless, we make use of the fixed-feature task transfer setting proposed in [7]. The recipe just slightly changes: split the data into $A$ and $B$ as usual and perform SAT. Fix all features, replace the last classification layer with a $1$-hidden layered classifier and finetune only the new classifier on the target task. Importantly, neither training nor validation set for the target task are split. We consider CIFAR-100 and ImageNet-200 and transfer to CIFAR-10, SVHN, Caltech-256 [35] and Flowers102 [36]. We call SAT trained for transfer Source-task Subset Adversarial Training (S-SAT), to emphasize that the subset training is performed on the source-task dataset.

In this section, we consider models that have "seen" only a fraction of AEs on the source task and investigate the robustness transfer capabilities to tasks on which they have not explicitly adversarially

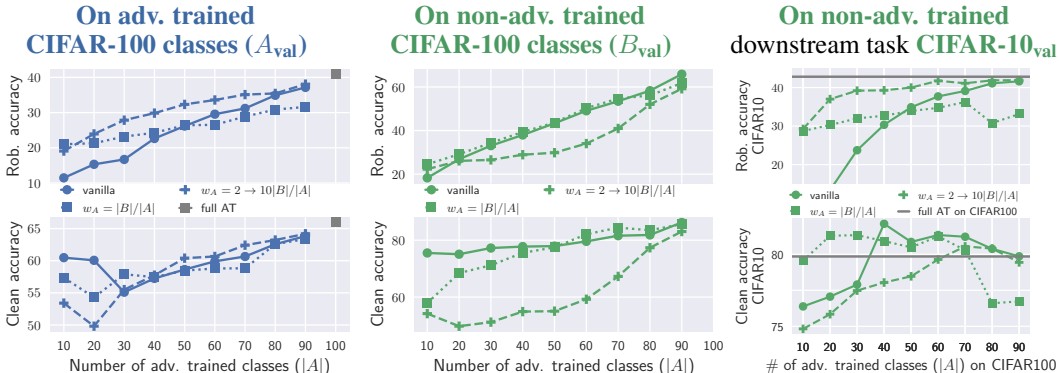

Figure 7: Impact of cross-entropy weighting on robustness transfer. For subset AT, we test different weighting strategies for sets A and B given they are of unequal size. We observe that vanilla cross-entropy (*circle*) offers the worst robustness transfer to CIFAR-10 (right). The best transfer (*plus*) is provided when loss weights are chosen such that training is overemphasized on A, indicated by dropping robust accuracies on B (compare left and center).

trained on. We find unexpectedly strong transfer performances for models that have both low clean and robust accuracy, only by putting more weight on the AEs.

**Loss balancing improves robustness transfer.** In contrast to the previously explored setting, we observe the transfer setting to benefit from loss balancing. Recall equation 2 in section 3.2 in which $w_A$ and $w_B$ can be assigned different values to balance the loss when $|A| \neq |B|$. We show that the vanilla configuration $w_A = w_B = 1$ transfers robustness to downstream tasks poorly, that balancing the loss with $w_B = 1, w_A = |B|/|A|$ lacks transfer performance for small $|B|$ and that weighting examples from $A$ higher results in improved robustness transfer. We present results for all three weightings in figure 7. The figure is organized in three columns, all reporting robust accuracy. The first column reports the robust accuracy on subset $A_{\text{val}}$, the second on subset $B_{\text{val}}$ and the third reports the robust accuracy on the downstream task. Here, we train on CIFAR-100 and transfer to CIFAR-10. The vanilla loss is indicated by circles and a solid line, the balanced loss $w_A = |B|/|A|$ by squares and a dotted line and the loss overemphasizing $A$ by a plus and a dashed line.

First and foremost, note that the robustness transfer for the vanilla configuration is substantially worse than both alternatives (robust accuracy in top right). Transfer improves with use of loss balancing, e.g. for $|A| = 10$, robust accuracy improves from $8\%$ to $30\%$, but does not converge to the baseline AT performance (gray line). This is an unwanted side effect of equalizing the weight between $A$ and $B$. When $A$ is much smaller than $B$, less weight is assigned to the AEs constructed for $A$ and robustness reduces. Note, this effect can also be seen on $A_{\text{val}}$ (top left in figure). Instead, we find it beneficial to overemphasize on the AEs (plus with dashed line). This configuration assigns $w_A = 2|B|/|A|$ for $|A| = 10$ and increases the weight to $w_A = 10|B|/|A|$ for $|A| = 90$. This results in improved robust accuray on $A_{\text{val}}$, but low robust and clean accuracy on $B_{\text{val}}$. Interestingly, while the generalization to $B_{\text{val}}$ is low, robustness transfer to CIFAR-10 is very high. We use this loss weighting for all following task transfer experiments.

**Robustness transfer from example subsets.** Using the weighted loss, we focus in the following on S-ESAT on two source tasks: CIFAR-100 and ImageNet-200, and train on three downstream tasks. Similar results for S-CSAT and SVHN as additional downstream task can be found in the supplement. Figure 8 presents results for three settings: CIFAR-100 $\rightarrow$ CIFAR-10 and ImageNet-200 $\rightarrow$ Caltech-256, Oxford-Flowers-102. The first and second row show robust and clean accuracy on the downstream task respectively. As before, we compare with a random (red) and a full AT baseline (gray line). Selecting $A$ to contain the hardest examples first (highest entropy) is marked by a solid line; selecting easiest is marked by a dashed line.

In line with the improvements seen using the appropriate loss weighting, we see similarly fast recovery of baseline AT performance across all dataset. In fact, $|A|$ containing only $30\%$ of training data (15k and 70k) is sufficient to reach near baseline performance. On CIFAR-100 $\rightarrow$ CIFAR-10 and ImageNet-200 $\rightarrow$ Flowers-102 even slightly outperforming the same with a further increase in size. Similar to the non-transfer settings tested before, we also see similar interactions between

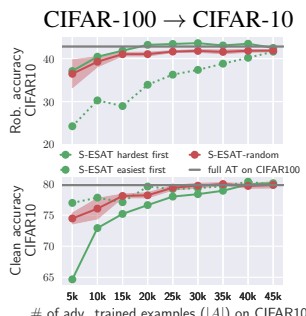
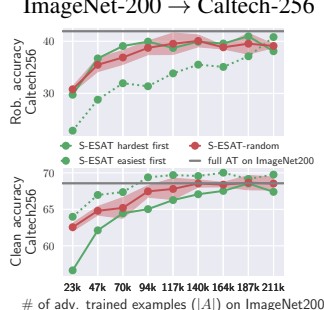
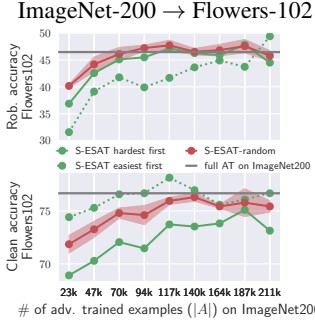

Figure 8: Transfer from S-ESAT to three different downstream tasks. S-ESAT is trained on source dataset CIFAR-100 (left) and ImageNet-200 (middle and right). We report robust (top row) and clean (bottom row) accuracies for increasing size of $A$. Similar to our investigation on transfer from $A$ to $B$, we find that hard examples provide better robustness transfer than easy ones, but random selections (red) achieve competitive performances. Most importantly, "seeing" only few AEs (here 30% of source data) recovers baseline AT performance (gray line).

subset selection strategies. I.e. hardest examples (solid line) provide greater robustness transfer than easiest (dashed line) while a random baseline (red) achieves competitive performances. The latter consistently outperforming entropy selection on ImageNet-200 → Flowers-102, supporting our observation in section 4.2: with increasing dataset complexity, informed subset selection provides diminishing returns. Note that all robust accuracy increases proportionally correlate to an increase in clean accuracy as well. This is in stark contrast to the inverse relationship in previous settings. C.f. figure 5 and 6, for which clean accuracy decreases. This interaction during transfer is similar to what is reported in [8]: increased robustness of the source model results in increased clean accuracy on the target task (over a non-robust model). Intriguingly though, with appropriate weighting, the biggest robustness gains on the downstream task happen under fairly small $A$. This is a promising outlook for introducing robustness in the foundational setting [37], where models are generally trained on very large datasets, for which AT is multiple factors more expensive to train. Note that our results generalize to single-step attacks like fast gradient sign method (FGSM) [18, 38] as well. We provide evaluations in the supplement. While we consider the fixed-feature transfer only, recent work has shown this to be a reliable indicator for utility on full-network transfer [8, 39].

## 5 Conclusion

In this paper, we presented an analysis of how adversarial robustness transfers between classes, examples and tasks. To this end, we proposed the use of Subset Adversarial Training (SAT), which splits the training data into $A$ and $B$ and constructs AEs on $A$ only. Trained on CIFAR-10, CIFAR-100 and ImageNet-200, SAT revealed a surprising generalizability of robustness between subsets, which we found to be based on the following observations: (i) adv. robustness transfers among classes even if some or most classes have never been attacked during training and (ii) hard classes and examples provide better robustness transfer than easy ones. These observations remained largely valid in the transfer to downstream tasks like Flowers-102 and Caltech-256 for which we found that overemphasizing loss minimization of AEs in $A$ provided fast convergence to baseline AT robust accuracies, even though transfer to $B$ was severely reduced. Specifically, it appears that only few AEs ($A$ containing 30% of the training set) learn all of the robust features which generalize to downstream tasks. This finding could be particularly interesting for AT in the foundational setting, in which very large datasets render training computationally demanding.

More broadly, improving adversarial robustness remains one of the most important problems to solve in deep learning, especially in high-stake decision making like autonomous driving or medical diagnostics. Our findings shed new light onto the properties of adversarial training and may lead to more efficient robustness transfer approaches which would allow easier deployment of robust models. We provided an account on a broad variety of datasets and used models commonly evaluated in related work. It needs to be seen whether our findings generalize to other threat models [40] as well.

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
