| Dataset | Classes | Size (Train/Test) |
|---|---|---|
| CIFAR-10 [30] | 10 | 50 000 / 10 000 |
| CIFAR-100 [30] | 100 | 50 000 / 10 000 |
| ImageNet-200 [32, 31] | 200 | 259 906 / 10 000 |
| Caltech-256 [35] | 257 | 24 485 / 6122 |
| Flowers-102 [36] | 102 | 1020 / 1020 |
| SVHN [42] | 10 | 73 257 / 26 032 |

Table 1: Number of training and validation examples per dataset used. ImageNet-200 uses examples from [32] only for classes defined in [31]

| Conventional setting | | | | | | |
|---|---|---|---|---|---|---|
| **Dataset** | **Architecture** | **Epochs** | **Batchsize** | **lr** | **lr-decays** | $L_2$ **decay** |
| CIFAR-10 | PreActResNet-18 | 200 | 128 | 0.1 | $100, 150$ | $5 \cdot 10^{-4}$ |
| CIFAR-100 | PreActResNet-18 | 200 | 128 | 0.1 | $100, 150$ | $5 \cdot 10^{-4}$ |
| ImageNet-200 | ResNet-50 | 150 | 256 | 0.1 | $50, 100$ | $1 \cdot 10^{-4}$ |
| Transfer setting | | | | | | |
| **Dataset** | **Architecture** | **Epochs** | **Batchsize** | **lr** | **lr-decays** | $L_2$ **decay** |
| CIFAR-10 | PreActResNet-18 + [512,10] | 40 | 128 | 0.1 | $20, 30$ | $5 \cdot 10^{-4}$ |
| SVHN | PreActResNet-18 + [512,10] | 40 | 128 | 0.1 | $20, 30$ | $5 \cdot 10^{-4}$ |
| Caltech-256 | ResNet-50 + [2048,257] | 100 | 128 | 0.1 | $50, 75$ | $1 \cdot 10^{-4}$ |
| Flowers-102 | ResNet-50 + [2048,102] | 100 | 102 | 0.1 | $50, 75$ | $1 \cdot 10^{-4}$ |

Table 2: Training settings for all used dataset for the conventional (upper rows) and the transfer setting (lower rows). In the transfer setting, the last classifier layer is replaced with two linear layers of size $K \times K$ and $K \times N$, abbreviated as $[K, N]$. $K$ defines the number of feature channels and $N$ the number of classes.

# A    Appendix

## A.1    Full training details

For all training setups listed in table 2, we train our models from scratch using SGD with a momentum of $0.9$. Dataset sizes are listed in table 1. All are data augmented based on the definitions in [41]. The sequence of transformations are listed in figure 9. Left, for CIFAR-10, CIFAR-100 and SVHN. Right, for ImageNet-200, Caltech-256 and Flowers-102.

**Adversarial training** is performed with 7 steps of projected gradient descent (PGD-7) within an $\epsilon = 0.5$ for CIFAR and SVHN and $\epsilon = 3.0$ for ImageNet-200, Caltech-256 and Flowers-102. For each step, we use a step size of $0.1$ and $0.5$ respectively. For all experiments, we maximize the default cross-entropy loss.

**Class order.** In the following, we list the order of classes ranked by entropy $\overline{\mathcal{H}}_C$ (equation 3). CIFAR-10 can be derived from figure 4 in the main paper. In figures 10 and 11, we provide the list for CIFAR-100 and ImageNet-200. On CIFAR-100, the first and thus hardest classes consist mostly of animate categories like *otter*, *rabbit* and *crocodile*. The easiest on the other hand are inanimate categories, specifically vehicle related classes, e.g. *road*, *motorcycle* or *pickup-truck*. Overall, the

```
 - pad 4 pixels
 - random crop to 32x32              - random crop to 224x224
 - random horizontal flip            - random horizontal flip
 - color jitter [0.25, 0.25, 0.25]   - color jitter [0.1, 0.1, 0.1]
 - random rotation within +/- 2 deg. - random rotation within +/- 2 deg.
```

Figure 9: Input transformation for CIFAR and SVHN datasets (left) and ImageNet-200, Caltech-256 and Flowers-102 (right) during training. During testing, no transformations are applied to CIFAR and SVHN. The remaining datasets are resized such that the shortest side equals 256, after which they are center cropped to 224.

```
otter, lizard, seal, rabbit, mouse, crocodile, lobster, shrew, shark,
woman, beaver, bowl, turtle, squirrel, possum, snail, girl, kangaroo,
ray, forest, caterpillar, man, baby, dinosaur, lamp, elephant, couch, boy,
porcupine, snake, butterfly, leopard, crab, table, mushroom, dolphin,
willow_tree, beetle, spider, clock, fox, sweet_pepper, bee, house,
raccoon, tulip, bridge, bus, rose, tank, whale, train, worm, lion, poppy,
trout, bed, plate, can, telephone, tiger, hamster, aquarium_fish,
maple_tree, orchid, pear, mountain, tractor, oak_tree, rocket, skunk,
cockroach, television, cup, sea, cloud, lawn_mower, castle, bottle,
palm_tree, keyboard, apple, plain, pickup_truck, bicycle, orange, chair,
wardrobe, motorcycle, road
```

Figure 10: CIFAR-100 classes ranked by decreasing entropy $\overline{\mathcal{H}}_C$. Animal classes are hardest, inanimate classes easiest.

```
spatula, shovel, syringe, drumstick, hand blower, lighter, nail, maraca,
barrow, umbrella, bow, quill, iron, stethoscope, soap dispenser, dumbbell,
mask, reel, toaster, ant, walking stick, envelope, candle, sleeping bag,
sandal, tricycle, cowboy boot, cradle, breastplate, bubble, banjo, chest,
cliff, wine bottle, fountain, crayfish, doormat, Chihuahua, chain, apron,
kimono, cockroach, accordion, sewing machine, ocarina, revolver, torch,
piggy bank, goblet, studio couch, wreck, hermit crab, grand piano, beaker,
snail, marimba, sundial, mantis, vulture, sea lion, flagpole, washer,
acoustic guitar, mongoose, grasshopper, Christmas stocking, bikini, corn,
balance beam, fox squirrel, American alligator, academic gown,
feather boa, suspension bridge, stingray, acorn, common iguana, forklift,
parachute, mushroom, hotdog, American black bear, beacon, garbage truck,
cello, pug, bee, banana, volcano, baboon, centipede, golfcart, marmot,
limousine, African chameleon, leafhopper, canoe, wood rabbit, agama,
starfish, lynx, German shepherd, capuchin, balloon, goose,
submarine, golden retriever, mitten, jeep, hummingbird, armadillo,
weevil, porcupine, puck, snowplow, barn, fly, tarantula, Rottweiler,
pool table, red fox, harvestman, pretzel, ballplayer, American egret,
puffer, ladybug, pelican, obelisk, bald eagle, go-kart, bell pepper,
castle, snowmobile, junco, lemon, spider web, lion, water tower,
basketball, guacamole, toucan, tank, jellyfish, viaduct,
robin, ambulance, broccoli, flatworm, pomegranate, bison, sea anemone,
jay, rugby ball, organ, drake, cheeseburger, mosque, koala, garter snake,
African elephant, lycaenid, oystercatcher, box turtle, cabbage butterfly,
steam locomotive, goldfinch, jack-o'-lantern, school bus, lorikeet,
manhole cover, rapeseed, flamingo, yellow lady's slipper, monarch
```

Figure 11: ImageNet-200 classes ranked by decreasing entropy $\overline{\mathcal{H}}_C$. In contrast to the order on CIFAR-10 and CIFAR-100, animate classes are generally not the most frequent among the hardest. Instead its mostly inanimate objects.

animate-inanimate order is similar to CIFAR-10. On ImageNet-200, we observe a very different order. Inanimate categories like *spatula*, *drumstick* or *umbrella* are among the hardest, while animate classes like *monarch (butterfly)*, *flamingo* or *lorikeet* are among the easiest. Named hard classes may be difficult to distinguish due to a frequent presence of people in the images.

## A.2 Alternative rankings

For simplicity, we focused our experiments on using entropy as a proxy to measure example and class difficulty (c.f. equation 3). Multiple such difficulty metrics have been proposed in literature [43, 16, 26, 27], of which we select a few from recent literature to compare to: signed variance (SVar) [16] and variance of gradients (VoG) [27]. We want to highlight, that they perform very similar to our entropy metric when utilized in our SAT framework. Figure 12 compares these two metrics with our used entropy metric using ESAT on CIFAR-100. Overall, VoG has a slight edge over SVar and Entropy,

434 yet the differences remain small. On $5k$ attacked examples, Entropy (yellow line) achieves $21.0\%$,
435 VoG (red line) $21.9\%$ and SVar (purple line) $22.3\%$ robust accuracy. On $25k$ attacked examples,
436 Entropy achieves $38.0\%$, VoG $38.8\%$ and SVar $38.1\%$. While some improvements over our simple
437 Entropy metric are possible, no proposed metric has a clear edge over the other.

## A.3   Full results for CSAT

439 Results for CSAT can be plotted for three different validation
440 subsets: $A_{val}$, $B_{val}$ and on the whole dataset $\mathcal{D}_{val}$. For clarity, we
441 only showed robust accuracies on $\mathcal{D}_{val}$ and $B_{val}$ in the main paper
442 in figure 5. Here, we provide all results. That is, in figure 13, we
443 show robust accuracies in the upper split and clean accuracies
444 in the lower split for all 3 subsets.

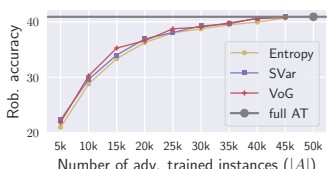

Figure 12: Various hardness metrics result in similar rob. accs. for ESAT on CIFAR-100.

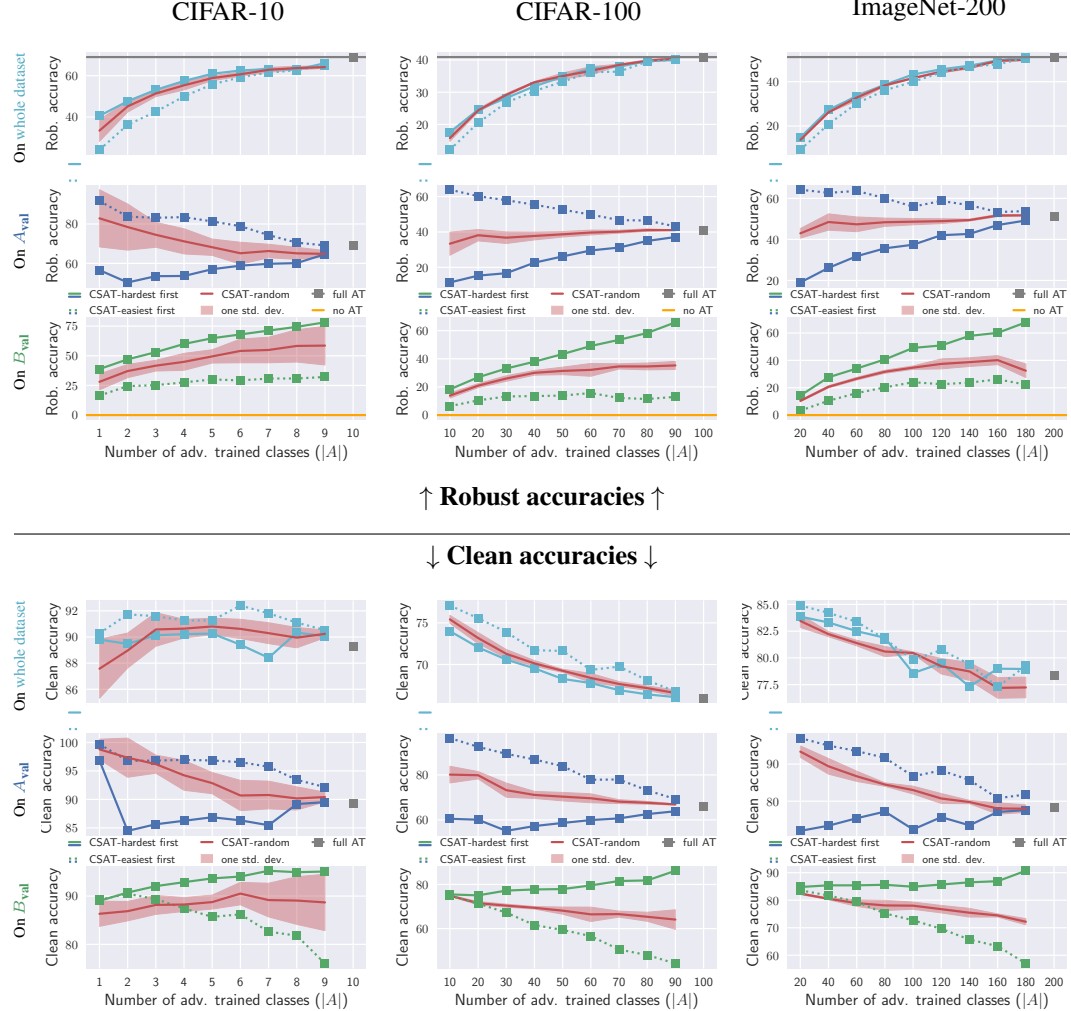

Figure 13: Full robust (upper split) and clean accuracies (lower split) from CSAT experiments, plotted for the whole dataset, $A_{val}$ and $B_{val}$. Selecting the hardest classes first (solid lines), clean accuracies and robust accuracies on $A_{val}$ steadily increase, while selecting the easiest in contrast (dotted lines) results in a steady decline. This provides additional support that entropy as metric provides a useful account of difficulty, since easy classes can achieve higher accuracy. Furthermore, we note that clean accuracy on the whole dataset is increasing or mostly stable, while on other datasets it is steadily decreasing. This should be investigated further.

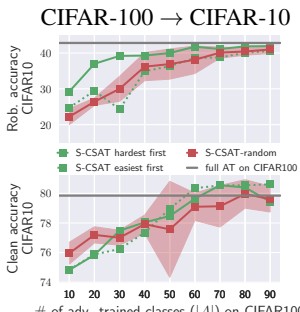
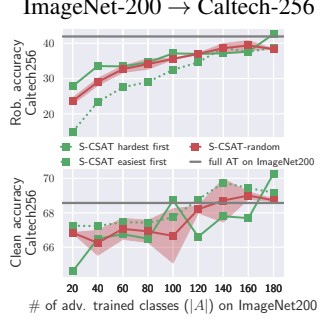
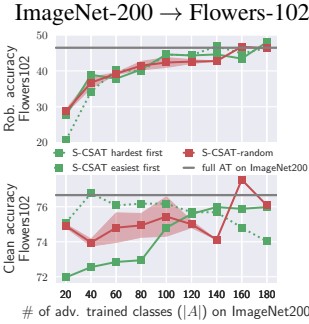

Figure 15: Transfer from S-CSAT to the same downstream tasks as in figure 8. S-CSAT is trained on source dataset CIFAR-100 (left) and ImageNet-200 (middle and right). We report robust (top row) and clean (bottom) accuracies for increasing size of $A$. We observe similar properties to S-ESAT, yet find convergence to the baseline AT performance to be substantially slower; in line with our discussion on SAT in section 3.2.

## A.4 Full results for transfer settings

In the main paper, we omitted transfer results to SVHN as well as using S-CSAT. Firstly, we provide the transfer result from CIFAR-100 to SVHN in figure 14. Robust accuracies are plotted on the upper plot, clean accuracies below. Note that $5k$ examples in $A$ are sufficient to reach baseline AT performance (gray line), while $15k$ provides a substantial improvement in robust accuracy ( $22\%$ vs $20\%$ ). Secondly, transfer results on S-CSAT aligned with the experiments in section 4.3 are shown in figure 15. We observe similar characteristics to the CSAT results in section 3.2, i.e. selecting the hardest classes first (solid line) is only advantageous on small $A$, while generally it draws even with the random baseline (red). Overall, convergence to the full AT baseline is slower than with S-ESAT.

## A.5 Single-step AT

While our main experiments use AT with 7 PGD-steps, we here show that non-trivial robustness transfer can be achieved with single-step AT as well. We focus on transfer to downstream tasks and compare with the results shown in figure 8, section 4.3. I.e., we train one ESAT model on CIFAR-100 and ImageNet-200 respectively, and finetune additional classifier on either CIFAR-10, Caltech256 or Flowers-102. We use *FGSM-RS* [38], with a step-size of 0.625 for $\epsilon = 0.5$ and 3.75 for $\epsilon = 3.0$. All other training settings are consistent with previous experiments (c.f. section A.1).

Results are shown in figure 16, comparing PGD-7 training (circles on solid line) and single-step FGSM-RS (squares on dotted line). Generally, we observe very similar clean and robust accuracies (lower and upper row) across all architectures. Specifically, FGSM-RS achieves slightly higher clean accuracies and slightly lower robust accuracies – especially for small $|A|$. Nonetheless, single-step AT converges to the full AT baseline (gray line) in a similar fast rate, i.e. generating AEs for around $30\%$ of the training set is sufficient.

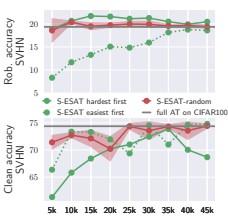

Figure 14: Robustness transfer from CIFAR-100 to SVHN using S-ESAT

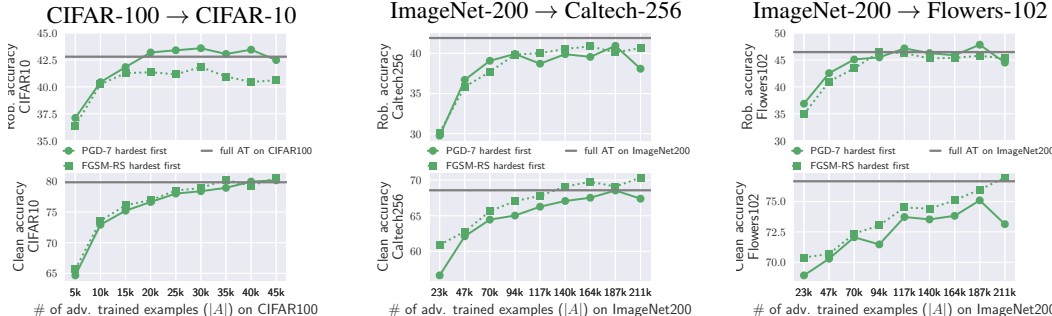

Figure 16: Comparison between PGD-7 and single step S-ESAT on the transfer setting to three different downstream tasks.