# OpenReview forum: "On Adversarial Training without Perturbing all Examples"
_NeurIPS.cc/2023/Conference — Submitted to NeurIPS 2023_

### Official Review · Reviewer_DBWu · 2023-07-01

**Soundness:** 4 excellent
**Presentation:** 4 excellent
**Contribution:** 2 fair
**Rating:** 7
**Confidence:** 4

**Summary:**

The authors propose a new approach called Subset Adversarial Training (SAT), which differs from traditional adversarial training methods that generate adversarial examples on the whole training set. Instead, SAT applies adversarial training on a subset of the training data. They studied two variants of subset adversarial training (CSAT and ESAT). They found that robust training in one class could generalize to other classes that were not adversarially trained, which is surprising. The also found that ESAT, where they adversarially trained on harder examples, gives surprising boost to downstream robust performance with much less data.

The paper also discusses the concept of loss balancing, which is used to counteract an imbalance between the adversarial subset and non-adversarial subset when the training split is not even. The authors found that loss balancing is important for the adversarial robustness transfer observed.

In conclusion, the paper presents a novel approach to adversarial training that could help us better understand the underlying mechanism of robust learning as well as having potential implication to more efficient adversarial training.

**Strengths:**

- The setting of experiment is interesting. It is surprising that adversarially training on a single class yields adversarial robustness to other classes. The originality of the experiment is strong.
- The experiments has demonstrated possibility to decrease the cost of adversarial training.
- The paper is also very clear with thorough experiments and analysis

**Weaknesses:**

- Even though the finding is interesting, I think the paper could do more in terms of understanding its implication to  robust generalization. What does robust generalization to other classes imply or show about the process of adversarial learning?
- While the experiment demonstrate possibility of decreasing cost of adversarial training, it doesn’t demonstrate this in more challenging scenarios. I understand that the paper’s intention is to understand how adversarial generalization happen as opposed to achieving the best performance, but if it is the paper’s intention to gain further understanding of the adversarial training process, I am hoping for more analysis and comments about its implication to robust generalization.

**Questions:**

One thing that I would like to see a little bit more of is analysis and discussion of the implication of the findings. For example, the authors have observed that the difficulty of a class is the main driver for its robust transferability to other classes. Could the author for example construct a synthetic 11th class on CIFAR-10 of varying difficulty that is completely unrelated to CIFAR-10, and see whether adversarial training on only this 11th class can give robust generalization to the original 10 classes. If the difficulty is increased, what is the extent of the robust generalization? I am a little bit unsure as to how the 11th class could be constructed, but this is an example of experiment where I would like to see the authors do by digging deeper into the possible implications.

**Limitations:**

The authors have adequately addressed the limitation of the potential negative societal impact of their work.

---

> ### Author Rebuttal · Authors · 2023-08-09
>
>  - **Q1: No comments on improving understanding of AT.**
>     While we agree that further insights into robustness transfer would be beneficial, we believe our submission remains of strong interest to the scientific community (note the reviewers comment on our suprising findings (R1, R2, R4, R5).
>     That is, our original experimental setup (R5) may act as basis for further research and lead to interesting findings down the road.
>
>  - **Q2: Suggested CIFAR-10+1 experiment.**
>     We appreciate the input for original experiments and have evaluated exactly such a setup.
>     We synthesized a set of 11th classes from CIFAR-100s super-classes -- of which there are 20, see below -- and perform CSAT on this 11th class to evaluate the robust accuracy gains on the original CIFAR-10 classes.
>     Results are reported in figure 3 in the rebuttal pdf.
>     We continue to observe a correspondence between average entropy $\overline{H}_c$ and the robustness transfer of a class.
>     As in the main paper, we evaluate $H_c$ on non-adv. trained models.
>     Note that, while the best performing setup ($A=$ *rodent*) with rob. acc of $33.3$\% does not improve upon the best in the main paper ($A=$ *cat*, with a rob. acc $> 37.8$\%), the number of examples in $A$ is only $|A|=2500$, thus less than $5$\% of training data.
>     This implies, that it could be possible to augment datasets with additional classes/examples that provide high $\overline{H}$ which in turn increase robust accuracy quickest. The result being: baseline AT performance with SAT on a very small but highly effective selection of $A$.
>     We will add this experiment and its implications to the final paper.
>
> CIFAR-100 Superclasses:
>
>     aquatic-mammal: ['beaver', 'dolphin', 'otter', 'seal', 'whale']
>     fish: ['aquarium_fish', 'flatfish', 'ray', 'shark', 'trout']
>     flower: ['orchid', 'poppy', 'rose', 'sunflower', 'tulip']
>     container: ['bottle', 'bowl', 'can', 'cup', 'plate']
>     fruit: ['apple', 'mushroom', 'orange', 'pear', 'sweet_pepper']
>     device: ['clock', 'keyboard', 'lamp', 'telephone', 'television']
>     furniture: ['bed', 'chair', 'couch', 'table', 'wardrobe']
>     insect: ['bee', 'beetle', 'butterfly', 'caterpillar', 'cockroach']
>     large-carnivore: ['bear', 'leopard', 'lion', 'tiger', 'wolf']
>     building: ['bridge', 'castle', 'house', 'road', 'skyscraper']
>     scene: ['cloud', 'forest', 'mountain', 'plain', 'sea']
>     large-mammal: ['camel', 'cattle', 'chimpanzee', 'elephant', 'kangaroo']
>     small-mammal: ['fox', 'porcupine', 'possum', 'raccoon', 'skunk']
>     crustacean: ['crab', 'lobster', 'snail', 'spider', 'worm']
>     human: ['baby', 'boy', 'girl', 'man', 'woman']
>     reptile: ['crocodile', 'dinosaur', 'lizard', 'snake', 'turtle']
>     rodent: ['hamster', 'mouse', 'rabbit', 'shrew', 'squirrel']
>     tree: ['maple_tree', 'oak_tree', 'palm_tree', 'pine_tree', 'willow_tree']
>     vehicle: ['bicycle', 'bus', 'motorcycle', 'pickup_truck', 'train']
>     utility-vehicle: ['lawn_mower', 'rocket', 'streetcar', 'tank', 'tractor']

---

> > ### Comment · Reviewer_DBWu · 2023-08-18
> > **Thank you for the additional experiments**
> >
> > Thank you for the additional experiment. I find the paper quite interesting, and have updated my score to accept.

---

### Official Review · Reviewer_mnhf · 2023-07-02

**Soundness:** 3 good
**Presentation:** 2 fair
**Contribution:** 2 fair
**Rating:** 5
**Confidence:** 4

**Summary:**

This paper demonstrates an interesting observation: when we conduct adversarial training, we can only choose to generate adversarial examples on a subset of the training data, if this subset contains the hardest examples, then adversarial training on a subset can achieve competitive performance in robustness over the whole dataset. In addition, models trained in such a manner demonstrate good transferable feature.

**Strengths:**

1. The observation is surprising and interesting, which indicates the transferability of adversarial examples across different classes.

2. The authors conduct comprehensive experiments on various datasets to validate the findings.

**Weaknesses:**

1. One major concern is the contribution, the proposed method neither improve the robust accuracy (or clean accuracy) nor improve the training efficiency (because SAT still uses PGD-7 to generate adversarial examples, which is inefficient).

2. All the experiments are conducted on the $l_2$ bounded adversarial perturbations, more types of adversarial perturbations should be included, especially the $l_\infty$ bounded ones which is popular for benchmarking. In addition, for CIFAR10, the adversarial budget $\epsilon = 0.5$ is very small when considering the dimensionality of the input image. Experiments based on larger adversarial budgets should be included, e.g. $\epsilon = 2$ for CIFAR10.

3. Similar to the first point, the experiments does not demonstrate the advantages of the proposed method. In addition to adversarial training, is the method general and compatible to other popular robust learning method, such as TRADES? Is the observation the same in this context?

4. It would be better if the authors can provide some intuition or explanations for the observations in this paper.

**Questions:**

In addition to the concerns in the "weakness" part that I expect answers from the authors, I have minor questions as below:

1. Why does the authors rank the difficulty of training instances based on the non-adv trained models, because the difficulty ranking can be quite different between adv and non-adv trained models.

2. Figure 8 demonstrates the transferability can vary a bit given the number of instances in the set A. For practitioners, does the authors have some hint for how to choose |A|?

Because of the weakness part and the questions in this section, I cannot recommend acceptance based on the current manuscript. I will do a re-evaluation after the authors' feedback.

**Limitations:**

The limitations and the broader societal impacts are not adequately discussed in the current manuscripts, although ethnicity should not be an issue for general research like this work.

---

> ### Author Rebuttal · Authors · 2023-08-09
>
>  - **Q1: Marginal contribution.** We reiterate our answer to Q4 of R3, but also add, that SAT works with 1-step FGSM-RS as well (see figure 14 in the appendix): We believe that a paper does not need to provide improved performance nor efficiency, if it can provide a set of experiments that highlight surprising phenomena or counter-intuitive results that could lead to new research directions.
>     We categorize our submission as the latter and we observe 4 reviewers to agree (R1, R2, R5 and R4 themselves).
>     In contrast to related work on robustness transfer and AT efficiency (as discussed in related work and our method section), we discuss a surprising phenomenon within robustness transfer between classes and examples. In our eyes, the most surprising being, that SAT on a single class (e.g. *cat*) can provide better robustness transfer to seemingly unrelated classes (e.g. *truck*) than SAT on a related class would do (e.g. *car*). We kindly ask the reviewer to reconsider her/his expectations on a good scientific paper.
>
>  - **Q2: Larger margin widths and $L_\text{inf}$.** $\epsilon=0.5$ is the standard for $L_2$ AT on CIFAR-10 (e.g. see [C3]).
>     Irrespectively, we agree that different $\epsilon$ budgets is an interesting set of experiments, although we note that $\epsilon=2.0$ is larger than the smallest $L_2$ distance between CIFAR-10 classes.
>     Consequently, we evaluate for $\epsilon=0.25$ and $\epsilon=1.0$.
>     Find the results in the rebuttal pdf in figure 2 a-c.
>     We continue to observe non-trivial robustness transfer to B for all $\epsilon$ budgets, with diminishing returns for larger $\epsilon$.
>     Notably though, small $\epsilon$ provide very strong transfer, especially on hard classes.
>    W.r.t. $L_\text{inf}$, we concur that such an evaluation is interesting and will do so for the final version.
>     For this rebuttal, we have repeated the S-ESAT experiments for ImageNet-200 $\rightarrow$ Caltech-256 and Flowers-102 using the $L_\text{inf}$ norm with $\epsilon=8/255$.
>     Additionally, we repeated the ESAT experiments on ImageNet-200.
>     Find the results in the rebuttal pdf in figure 1 a-c.
>     We observe very similar characteristics as for $L_2$.
>
>  - **Q3: Practical advantages and compatibility with TRADES.**
>     Similar to our answer to Q1, we believe that scientific work does not necessitate having immediate practical applications.
>     In our case, we investigate robustness transfer and provide a series of unexpected results.
>     We anticipate, that our insight: robust features generalize surprisingly well to unseen classes and examples (especially for downstream tasks), will spur additional studies on making AT less data hungry.
>     Note that we discuss such a use case in section 4.3 with downstream task transfers. Here it is noteworthy, that it is sufficient to use only $30$\% of training data with S-ESAT and still achieve near baseline AT performance on the target task.
>     This is of particular interest in the foundational setting, where off-the-shelf AT models often don't exist.
>     Additionally, we highlight that SAT can be used to synthesise training sets: that is, add classes or examples providing high entropy $\overline{H}$ that in turn quickly increase robust accuracy.
>     That such a setup can work, is discussed in an additional experiment in the rebuttal pdf, figure 3. For a discussion, we kindly refer the reviewer to our response to R5 (DBwu).
>
>     TRADES is an adjusted loss to trade-off robustness for clean accuracy.
>     With that, we have no reason to believe that it provides conflicting results with SAT.
>     Nonetheless, investigating the degree of robustness transfer w.r.t. TRADES would be an interesting evaluation for future work, that we think is out of scope for this submission.
>
>  - **Q4: Missing intuitions.**
>     Precisely this question will likely lead to improved AT.
>     At this point, we cannot provide a resolution, but conjecture that our submission will excite other groups to pursue an answer to the phenomenon subject in our submission.
>
>  - **Q5: Ranking instances.**
>     We chose ranking instances on non-adv. trained models, as it is closer to existing work in literature (see section A.2 in the appendix).
>
>  - **Q6: Hint to choose $|A|$.**
>     We kindly ask the reviewer to be specific about the stated claim: ``transferability varies a lot''.,
>     That is, on the contrary, we observe consistently strong transfer when using random or hard example rankings.
>     It is to be expected, that robust accuracy is lowest when $|A|$ is low and otherwise high when $|A|$ is large.
>     What is unexpected though, is that the robust accuracy reaches near baseline AT performance when $|A|$ is only about $30$\% of the source task training data.
>     Based on this observation, it would make sense to give the following recommendation: for downstream tasks, around $30$\% of source data is sufficient -- ideally the hardest examples. To reach near baseline performances on the source task, around $50$\% of data is sufficient (again, ideally the hardest).
>
> [C3] Croce, Francesco, and Matthias Hein. "Reliable evaluation of adversarial robustness with an ensemble of diverse parameter-free attacks."ICML, 2020.

---

### Official Review · Reviewer_mJ8b · 2023-07-03

**Soundness:** 2 fair
**Presentation:** 2 fair
**Contribution:** 2 fair
**Rating:** 5
**Confidence:** 5

**Summary:**

The authors proposed the use of Subset Adversarial Training (SAT), a technique that splits the training data into A and B and constructs AEs only for data in A. Using SAT, they demonstrate how adversarial robustness transfers between classes, examples, and tasks. The authors report several insights: 1) that they've observed robustness transfers by difficulty and to classes in B 2) hard examples to provide better robustness transfer, and 3) Generating AEs on part of the data (e.g, 50%) is enough to get the standard AT accuracy.

**Strengths:**

1. The paper is relatively easy to follow
2. Existing empirical results seem sound

**Weaknesses:**

1. If I understand correctly, the experiments were done only on L2, even though the most common AT is done using L_inf. Can the authors present results using L_inf?
2. I'm missing many details about the AT process, you need to be much more specific for reproduction purposes. which AT is the baseline? did you try other methods? which method did you use? Madry's/TRADES/Other? the paper needs to be much clearer. Many important implementation details are missing.
3. It's hard to validate the results without supplying code/models.
4. The novelty is marginal, due to the fact that much prior art exists on the transferability of AEs, revisiting hard examples, or pruning a part of the training examples throughout the training. The paper will benefit from a comparison of these methods to SAT, so we can see the differences in performance/resources requirements/etc.

**Questions:**

See Weaknesses section.

**Limitations:**

No discussion on limitations, I suggest the authors to add one.

---

> ### Author Rebuttal · Authors · 2023-08-09
>
>  - **Q1: Evaluation on L\_inf.**
>     We concur, that evaluation on L\_inf is an interesting addition to our submission and will do so for the final version.
>     For this rebuttal, we have repeated the S-ESAT experiments for ImageNet-200 $\rightarrow$ Caltech-256 and Flowers-102 using the L\_inf norm with $\epsilon=8/255$.
>     Additionally, we repeated the ESAT experiments on ImageNet-200.
>     Find the results in the rebuttal pdf in figure 1 a-c.
>     We observe very similar characteristics as for L2.
>
>  - **Q2: Implementation details.**
>    All training details can be found in the second paragraph of section 4 and in the appendix. As stated in line 416-417, we use traditional adv. training optimizing the cross-entropy loss. Hence we do not use TRADES.
>
>  - **Q3: Unpublished code/models.**
>    We have compiled an anonymized repository here: https://anonymous.4open.science/r/SAT-BF9B/.
>     Model checkpoints for our main results will be made public with the final version.
>     We kindly ask the reviewer to specify, which model checkpoints they would like to have access to and we gladly provide them during the discussion phase.
>
>  - **Q4: Marginal contribution.**
>     We believe that a paper does not need to provide improved performance nor efficiency, if it can provide a set of experiments that highlight surprising phenomena or counter-intuitive results that could lead to new research directions.
>     We categorize our submission as the latter and we observe 4 reviewers to agree (R1, R2, R4, R5).
>     In contrast to related work on robustness transfer and AT efficiency (as discussed in related work and our method section), we reveal a surprising phenomenon within robustness transfer between classes and examples. In our opinion, the most surprising being, that SAT on a single class (e.g. *cat*) can provide better robustness transfer to seemingly unrelated classes (e.g. *truck*) than SAT on a related class would do (e.g. *car*).
>     Along these lines, we kindly ask the reviewer to consider what our submission can contribute to the research community.
>
>  - **No limitations.**
>    Please note, that we have added a limitation and broader impact statement in the conclusion section, as is accepted according to NeurIPS author guidelines.

---

> > ### Comment · Reviewer_mJ8b · 2023-08-19
> >
> > I thank the authors for their additional resources and experiment. I've raised my score to 5.

---

### Official Review · Reviewer_NsHp · 2023-07-11

**Soundness:** 2 fair
**Presentation:** 3 good
**Contribution:** 3 good
**Rating:** 7
**Confidence:** 4

**Summary:**

This work considers the transferability of adversarial robustness for partially adversarially trained models. The authors examine 3 variants of subset adversarial training (SAT): Class SAT, where only samples from selected, difficult classes are adversarially perturbed in training; Example SAT, where only examples with the highest predictive entropy are perturbed; Source-task SAT, where SAT-trained, robust models are fine-tuned on downstream training sets and evaluated for downstream adversarial robustness. They further draw connections between SAT and loss balancing, thus proposing a method for sample-efficient, low-cost adversarial robustness transfer between datasets in foundational settings.

The authors report interesting insights from various experiments. From CSAT, it is noted that difficult classes transfer best; class-wise transfer gains are asymmetric; and robustness transfers between seemingly unrelated classes. From ESAT, the authors concur with previous findings that harder examples contribute more to training robust models; the gain in robust accuracy is more rapid than CSAT with respect to the size of subset A; hardness rankings suffer from a possible lack of sample diversity and its performance is matched by random rankings. From S-SAT, they find that SAT on the source dataset with only 30% of AEs can match the robustness transfer gains using normal adversarial training, on the downstream dataset; both clean and robust downstream accuracies are transferred and they are positively correlated under appropriate loss balancing.

**Strengths:**

1. **Data efficiency.** The proposed SAT greatly reduces the amount of data required for adversarial training, which is promising for resource limited or real-world settings. ESAT with only 50% of AEs matches normal AT performance; S-SAT with 30% of AEs matches AT (on the source dataset) as well.
2. **Loss balancing.** I appreciate the discussion on connections between the SAT formulation and loss balancing. I also recognise that both clean and robust accuracy transfer positively  from source to downstream tasks, under S-SAT with appropriate loss balancing, which is rare and difficult for adversarial training.
3. **Experimentation.** The experiments are relatively thorough (except that only ResNet-18 and ResNet-50 are SAT-trained) and many details (such as the inter-class robustness transfer statistics for CSAT, or the difficulty rankings of classes) are provided, which give rise to valuable insights.
4. **Presentation.** The presentation of this work is exemplary. It is well-organised, logically-coherent and persuasive.

**Weaknesses:**

### 1 Cost and efficiency
1.1 $\hspace{5pt}$ SAT relies on meticulous pre-processing to discover hard classes and requires access to the per-epoch model weight snapshots of a normally trained classifier, to compute the difficulty metric of Equation 3. This shifts the partial cost of adversarial training to the pre-processing stage and can be costly for large models / datasets in foundational settings.
1.2 $\hspace{5pt}$ More importantly, SAT relies on this non-robust classifier, presumably with identical architecture and training data as the target model for subset adversarial training. This means that all the pre-training and loss balancing procedures have to be repeated for every single new model-dataset combination, which might end up being more costly than normal adversarial training.
1.3 $\hspace{5pt}$ One also notes that hardness ranking is important for the guaranteed performance of SAT, especially for CSAT and to a lesser extent for ESAT (where the authors acknowledge that it is "possible to accidentally select poor performing subsets", as per the easy rankings experiment).
### 2 Experimentation
2.1 $\hspace{5pt}$ SAT is only verified for ResNet-18 and ResNet-50, which is a non-negligible shortcoming, for the reasons described above.
2.2 $\hspace{5pt}$ Does SAT hold for other convolutional and non-convolutional architectures of variable capacity?
2.3 $\hspace{5pt}$ Is it not cost and time prohibitive to run SAT for more than 2 baseline models? Would this also be a barrier that impedes the practical adoption of SAT?
### 3 Scaling up
3.1 $\hspace{5pt}$ SAT experiments are notably performed on smaller datasets with fewer (or a subset of) classes. The computational complexity of SAT seems to scale non-negligibly with the number of classes (CSAT) and the size of the dataset (ESAT), which is not ideal on real-world datasets with fine-grained labels in foundational settings.
3.2 $\hspace{5pt}$ As aforementioned, even considering the S-SAT setup (where one does not need to do SAT on every downstream dataset), it is very costly to add a new model to the SAT experiments because of the method's dependence on snapshots of a normally-trained, non-robust version of the same model, for hardness rankings and loss balancing.

**Questions:**

### Major concerns.
1. Could the authors address the concern about shifting the time/computational cost of adversarial training from the training phase to pre-training / pre-processing phases?
2. Furthermore, even if the non-robust, pre-trained weight snapshots of the target model are available off-the-shelf, using S-SAT still further requires one to adversarially pre-train from scratch a model on a large-scale dataset. Since the transferred robustness of S-SAT is comparable to that of normally AT'd models on the source dataset, could the authors elucidate what is the particular advantage of SAT (as opposed to simply fine-tuning with an off-the-shelf robust, AT'd model)?
3. Could the authors suggest an efficient method of validating SAT for different dataset-model combinations (which does not require recomputation of task-specific hardness rankings and loss balancing settings); or alternatively, validate SAT on diverse convolutional and non-convolutional baselines?
### Minor comments.
4. What do the authors think about the connections between loss balancing and oversampling? How does CSAT perform for long-tailed or imbalanced datasets?
5. Have the authors considered the impact of homogeneous and heterogeneous data splitting methods (and resultant label or covariate shifts) on SAT models' clean and robust accuracy?

**Limitations:**

The authors have adequately addressed the societal and ethical limitations of their work. This work strives to improve the foundational adversarial robustness of AI systems in practice; experimental and implementation details have been documented.

---

> ### Author Rebuttal · Authors · 2023-08-09
>
>  - **Q1 Cost and Efficiency: SAT relies on meticulous pre-processing.**
>     Given a new architecture and a new dataset, the experimental process of finding good performing configurations involves training multiple models. During training, the entropy statistics for SAT can be cheaply computed without any substantial overhead, since softmax is applied to every instance anyway. Consequently, if SAT is to be adopted, such statistics can be computed on the fly.
>
>  - **Q2 Experimentation: Why use S-SAT over finetuning off-the-shelf adv. trained model.**
>     Our assumption w.r.t. foundational models is, that off-the-shelf adv. trained models do not exist since they are prohibitively expensive to train. In such a case, we'd argue that utilizing S-SAT can reduce the computational complexity down to 30\% and still reach competitive adv. robustness on downstream tasks.
>
>  - **Q2 Experimentation: SAT is only verified on ResNet-18 and ResNet-50.**
>     First and foremost, we reiterate, that our study is not on improving AT efficiency, but on revealing
>     To show this phenomenon generalizes to other architectures, we trained and evaluated SAT for $\epsilon_2 = 0.5$ with $A$ containing 50\% of data on wide-resnet with width 16 and depth 70, as is common in literature on AT[C1, C2].
>     Our baseline WRN-70-16 achieves a clean accuracy of 83.8\% and robust accuracy of 62.1\%.
>     Our ESAT achieves a clean accuracy of 84.8\% and a robust accuracy of 57.0\%.
>     These results are in line with our main paper observations.
>     Code for this new experiment can be found here: https://anonymous.4open.science/r/SAT-BF9B/ .
>
>  - **Q2.3: Cost prohibitive baseline training?**
>     We have shown that 50\% of training data recovers baseline performance for ESAT and 30\% of data for S-ESAT. If adopted, no baseline adversarial training is needed in practice -- only if needed for sanity check or comparison. Clean baselines (without AT) on the other hand, remains necessary to determine the entropy ranking.
>
>  - **Q3 Scaling: SAT is difficult to scale.**
>     While the full range of experiments is costly to scale, the point of SAT is not to provide a definite recipe for improving AT. Instead, SAT provides a means to investigate facets of AT.
>
>  - **Q4 and Q5 Imbalanced datasets.**
>     Thank you for providing two very interesting avenues for future work. Given our results on robustness transfer and loss balancing, it is indeed plausible to assume that even an undersampled and hard class can contribute substantially to training robust features for the whole model. At this point though, it is difficult to make any strong predictions.
>
> [C1] Rebuffi, Sylvestre-Alvise, et al. "Fixing data augmentation to improve adversarial robustness." arXiv preprint arXiv:2103.01946 (2021).
>
> [C2] Gowal, Sven, et al. "Improving robustness using generated data." NeurIPS (2021).

---

> > ### Comment · Reviewer_NsHp · 2023-08-20
> > **Thank you for the Clarifications**
> >
> > I have read the authors' rebuttal and other reviews in detail. I maintain that the reasons for acceptance outweigh reasons for rejection. In light of other reviews, I have increased my score from 5 to 7. Although I still have reservations about the cost overhead of using SAT in practice, I believe SAT's findings regarding the transferability (across classes and hard examples) of adversarial training to be fresh and noteworthy. I thank the authors for clarifying that the main objective of this work is "not to provide a definite recipe for improving AT" but rather to provide "a means to investigate facets of AT". I look forward to future work that further connects robustness transfer, loss balancing and sampling.

---

### Official Review · Reviewer_8YTs · 2023-07-13

**Soundness:** 3 good
**Presentation:** 3 good
**Contribution:** 2 fair
**Rating:** 5
**Confidence:** 3

**Summary:**

This paper investigates the transferability of adversarial robustness among different classes and different examples. Different from previous studies, authors split the training dataset into two groups and only apply adversarial training on one group while another one using clean training. Based on experiment results, authors obtain several interesting observations, including classes without adversarial training can still have some capacity to defense against adversarial attacks, hard classes and examples can provide better robustness transferability than easier ones, and only 50% of training data is sufficient to recover the performance with vanilla adversarial training method in terms of robustness, etc.

**Strengths:**

1. This paper explores the transferability of adversarial robustness from a new perspective and hence proposes a novel training mechanism to study.
2. This paper conducts a series of experiments to investigate the robustness transferability. Based on experiments, some interesting observations are obtained, which may give some new insights for future works.

**Weaknesses:**

1. The motivation of this work is not quite clear. Although authors find that some classes still can obtain capacity to defense against adversarial attacks without adversarial training, data samples of these classes are available in the training dataset. Hence, applying adversarial training on all data samples of all classes directly can achieve much better robustness, comparing with the transferred robustness obtained in this paper. Hence, it's not clear why authors study this kind of transferability when all training data are available and which scenarios are suitable for the problem studied in this paper.
2. Based on experimental results, authors claim that utilizing only half training data can achieve comparable robustness performance with vanilla adversarial training methods. However, related experiments only report robustness of different methods. Considering the trade-off between clean accuracy and robustness in adversarially trained models, it would be better if corresponding clean accuracy of each method can also be provided.

**Questions:**

1. Can authors discuss more about possible application scenarios or benefits of the study conducted in this paper? Although the transferability of adversarial robustness between different classes and examples under different training mechanisms is observed in this paper, it seems this observation cannot bring benefits to real applications. In practice, simply applying adversarial training on all data samples of all classes can achieve much better robustness. Hence, it‘s not clear what's the benefits of the study conduced in this paper in real applications, considering all data samples of all classes are used in training stage.
2. Can authors provide some understanding about why the rank class split cannot consistently outperform the random class split on CIFAR10 validation set in terms of both clean accuracy and robustness as shown in Figure 4? Does this result indicate the hardness based split on class level cannot boost model overall performance?

**Limitations:**

Questions about how to apply the proposed training mechanism and observations obtained from experiments in real applications to boost the robustness of models need to be discussed in detail.

---

> ### Author Rebuttal · Authors · 2023-08-09
>
>  - **Q1: Motivation not entirely clear. How could SAT be useful in real applications?**
>     We emphasize, that our study is not one of improving existing methods, but of improving our understanding of adversarial training (AT) and its robustness transfer.
>     In that, we find our observations to be of high interest to the community -- as R2, R4, R5 and R1 (theirself) noted.
>     We anticipate, that our insight: robust features generalize surprisingly well to unseen classes and examples (especially for downstream tasks), will spur additional studies on making AT less data hungry.
>     Note that we discuss such a use case in section 4.3 with downstream task transfers. Here it is noteworthy, that is sufficient to use only $30$\% of training data with S-ESAT and still achieve near baseline AT performance on the target task.
>     This is of particular interest in the the foundational setting, where off-the-shelf AT models often don't exist.
>     Additionally, we highlight that SAT can be used to synthesise training sets: that is, add classes or examples providing high entropy $\overline{H}$ that in turn quickly increase robust accuracy.
>     That such a setup can work, is discussed in an additional experiment in the rebuttal pdf, figure 3. For a discussion, we kindly refer the reviewer to our response to R5 (DBwu).
>
>  - **Q2: Detailed comparison of SAT impact on clean accuracy.**
>     Please note, that all experiments starting from figure 4 contain clean accuracies.
>     Figure 5 (CSAT), reports clean accuracies in the appendix in figure 13.
>     To summarize: for all CSAT and ESAT experiments, we observe decreasing clean accuracy with increasing $|A|$ -- as is expected for adv. trained models. Interestingly, this is not the case for S-CSAT and S-ESAT: here we observe *increasing* clean accuracy with increasing $|A|$.
>
>  - **Q3: Why does the informed ranking in SAT converge to random rankings for large $|A|$ (e.g. fig. 4)?**
>     For all rankings, we observe convergence to the full AT baseline.
>     The difference here lies in how quickly each of these rankings achieve this convergence.
>     Here, the informed ranking can improve robust accuracy quickest with fewest training examples.
>     Overall, this advantage diminishes though with larger $|A|$.

---

> > ### Comment · Reviewer_8YTs · 2023-08-20
> > **Thank you for your clarifications**
> >
> > I have read authors clarifications and I think they have addressed all my concerns in my review. Hence, I raised my score to 5.

---

### Author Rebuttal · Authors · 2023-08-09

We thank all reviewers for their time and valuable feedback on our manuscript. We are very pleased to read that R1, R2, R4 and R5 found our insights interesting, surprising (R4, R5) and that it may provide insights for future works (R1).
Our experiments were commented to be clear and thorough (R3, R5) and comprehensive (R4).
Furthermore, we highlight that our presentation was found to be exemplary and persuasive by R2 and was rated excellent by R5.
We address each reviewer in individual comments and provide an additional pdf with figures supplementing our rebuttal.

---

### Decision · Program_Chairs · 2023-09-21

**Decision:**

Reject

**Comment:**

The paper presents a novel approach to adversarial training by only perturbing a subset of examples. The work is innovative and has generally received positive feedback on its insights. However, there are notable concerns about the real-world applicability, cost-efficiency, and the comprehensiveness of evaluations. While the authors have attempted to address these concerns in their rebuttals, the issues raised are significant enough to warrant further revisions. The reviewers' concerns, especially about theoretical justifications and broader empirical evaluations, are valid and need to be addressed comprehensively.

Upon discussion, Reviewer mnhf maintains that the contribution of the paper is somewhat marginal, while Reviewer NsHp expresses reservations about the practical cost implications of implementing the proposed SAT approach. Unfortunately, no reviewer has stepped forward to champion the paper. After a careful examination of both the paper and the reviews, AC concurs that the concerns about real-world applicability, cost, and the comprehensiveness of evaluations are significant. The limited experimental evidence provided in the rebuttal does not sufficiently alleviate these concerns.

Given these considerations, the recommendation is to "Reject" the paper in its current form for this submission cycle. However, the authors are encouraged to undertake the necessary revisions and consider resubmitting in a future cycle.